# Implications of Protein and Sarcopenia in the Prognosis, Treatment, and Management of Metabolic Dysfunction-Associated Steatotic Liver Disease (MASLD)

**DOI:** 10.3390/nu16050658

**Published:** 2024-02-26

**Authors:** Avneet Singh, Adam Buckholz, Sonal Kumar, Carolyn Newberry

**Affiliations:** 1Department of Medicine, Cooper University Hospital, Camden, NJ 08103, USA; adulku470@gmail.com; 2Division of Gastroenterology, Weill Cornell Medical Center, New York, NY 10065, USA; apb9012@med.cornell.edu (A.B.); sok9028@med.cornell.edu (S.K.)

**Keywords:** fatty liver, metabolic dysfunction-associated steatotic liver disease, hepatic steatosis, sarcopenia, malnutrition

## Abstract

Metabolic Dysfunction-Associated Steatotic Liver Disease (MASLD) is a common cause of chronic liver disease globally, with prevalence rapidly increasing in parallel with rising rates of obesity and metabolic syndrome. MASLD is defined by the presence of excess fat in the liver, which may induce inflammatory changes and subsequent fibrosis in high-risk patients. Though MASLD occurs frequently, there is still no approved pharmacological treatment, and the mainstay of therapy remains lifestyle modification via dietary changes, enhancement of physical activity, and management of metabolic comorbidities. Most nutrition research and clinical guidance in this disease centers on the reduction in fructose and saturated fat in the diet, although the emerging literature suggests that protein supplementation is important and implicates muscle mass and sarcopenia in disease-related outcomes. This review will assess the current data on these topics, with the goal of defining best practices and identifying research gaps in care.

## 1. Introduction

Metabolic Dysfunction-Associated Steatotic Liver Disease (MASLD), previously named non-alcoholic fatty liver disease (NAFLD), is one of the most common etiologies of chronic liver disease worldwide, in addition to alcohol and viral-related hepatic disease [1]. Previously, NAFLD was diagnosed via imaging or histologic findings of hepatic steatosis without secondary causes of liver disease [2]. The new MASLD diagnostic criteria necessitate the presence of cardiometabolic disease risk factors, examples of which include elevated BMI, fasting serum glucose or hemoglobin A1C (HbA1c), blood pressure, or abnormal lipid profiles. The updated nomenclature also allows for the diagnosis of overlap syndromes, most notably patients with metabolic disease who also have risk factors for Alcoholic Liver Disease (ALD), a condition now known as Metabolic Alcoholic Liver Disease (MetALD). This recognition in dual etiology acknowledges the increased relative risk of liver complications in those with metabolic risk factors as well as moderate alcohol use, which is currently defined by the American Association for the Study of Liver Diseases (AASLD) as >140–350 g per week of alcohol for females and >210–420 g per week of alcohol for males [3]. Globally, the prevalence of MASLD is estimated to be 25–30% and 7.4% in adults and children, respectively, with an increase in diagnosis expected [4]. This rise is driven by the increasing prevalence of cardiometabolic risk factors such as obesity, insulin resistance, and hypertension, which correlates with enhanced intake of dietary fructose, fast food products, and inactivity [5,6]. MASLD has multiple manifestations. The first is basic steatosis, which can be present with or without inflammation. Significant inflammatory changes are consistent with Metabolic Dysfunction Associated with Steatohepatitis (MASH), which is characterized by hepatocellular injury and specific histological changes including lobular inflammation and hepatocyte ballooning on biopsy in the presence of steatosis. MASH with fibrosis indicates additional progression of disease defined by bridging fibrotic changes, which can ultimately progress to cirrhosis and end-stage liver disease [7].

### Diagnosis, Staging, and Management of MASLD

While the gold standard for diagnosis and staging of MASLD is liver biopsy, this is impractical for broad applicability given the overall prevalence of the condition and frequency of low-risk disease states without significant concern for progression and disease-related complications. Instead, imaging evidence (such as ultrasound) of steatosis is often combined with non-invasive risk stratification tools for predicting the degree of fibrosis and the need for additional assessment and intervention. Common risk stratification tools include serum-based scores, most commonly the FIB-4 score, and elastography-based tools including vibration-controlled transient elastography, or FibroScan, and MR Elastography [5,8,9].

Management algorithms for MASLD usually incorporate risk stratification serum-based scores, such as FIB-4, in addition to imaging and/or histological assessment. If risk-stratification scores confirm a high risk of advanced disease, imaging and/or histologic proof of steatosis is recommended to confirm diagnosis and assess disease activity. The FIB-4 index is a non-invasive tool that can be used as a screening tool to risk stratify for the probability of advanced fibrosis, which can, in turn, help with targeted interventions and appropriateness for referrals to hepatology providers. The most common imaging modality to assess for hepatic steatosis is ultrasonography and the gold standard for diagnosis is a liver biopsy. Other non-invasive imaging modalities that can be useful are transient elastography (i.e., Fibroscan) and MR elastography, which can calculate liver stiffness measurements and help with fibrosis scoring, as well as deciphering if steatosis is present [2,8]. Complications of MASLD include progression to MASH, fibrosis, and cirrhosis.

Fibrosis staging is important in patients with high-risk MASLD as there is an exponential growth in all-cause mortality in patients with increasing fibrosis stage [9]. In general, the most common cause of death among those with MASLD is cardiovascular disease, but those with significant hepatic fibrosis are at additional increased liver-related mortality [2]. In terms of liver-related complications, hepatocellular carcinoma (HCC) in MASLD can be present in patients with or without cirrhosis, but prevalence is increased in MASH cirrhosis, a population of which should follow standard surveillance imaging guidelines [5,10]. While there is no FDA-approved treatment for MASLD, there have been several studies showing the efficacy of antidiabetic drug classes, including thiazolidinediones and glucagon-like peptide-1 agonists (GLP1RAs), in improving liver histology and regressing steatosis and fibrosis. The first-line intervention, however, is still weight loss with diet and exercise in the absence of formal pharmacological approval. Studies have shown that a 7–10% total body weight loss improves MASH histology and is the target of lifestyle interventions and weight loss planning [2,11]. The literature assessing body compositional changes including fat and muscle mass is still in its infancy but is thought to be an important factor in disease-related activity and associated mortality.

## 2. Sarcopenia

The term sarcopenia was first used by Irwin Rosenberg in 1989, derived from the Greek terms ‘sarx’, which means flesh, and ‘penia’, meaning loss [12]. Originally, sarcopenia was defined as an age-related loss of lean body mass (LBM) [12]. In 2019, this definition was broadened and refined by the European Working Group on Sarcopenia in Older People (EWGSOP2) to denote a syndrome that involves both loss of skeletal muscle mass and muscle strength and function with associated adverse effects such as poor quality of life, increased frailty, and increased mortality [13].

In 2010, the EWGSOP formulated three criteria for sarcopenia diagnosis, which include the following: (1) low muscle mass; (2) low muscle strength; and/or (3) low physical performance (Table 1, Table 2 and Table 3) [14]. In 2019, EWGSOP2 emphasized that muscle strength should serve as the primary criteria for defining sarcopenia, with physical performance as a marker for assessing severity. Although determining muscle mass quantity and quality are necessary in diagnosing sarcopenia, their use as primary parameters can be limited in clinical practice. Diagnostic criteria have, in response, more recently been modified to include the presence of low muscle strength as probable sarcopenia and documentation of low muscle quality or quantity as confirmed sarcopenia. If low physical performance is also present, severe sarcopenia is diagnosed representing a combination of low muscle mass and/or function and resultant clinical limitations to individual physical endurance and strength [13].

Initially considered a disease confined to the elderly, several conditions are now known to considerably enhance sarcopenia risk, including states of prolonged physical inactivity, nutritional conditions that both limit oral calories and protein and/or reduce absorption of these nutrients, dysregulated insulin pathways, androgen deprivation states, wasting liver and kidney diseases, and malignancy [15,16]. Studies have shown that sarcopenia in advanced liver disease is associated with an increase in all-cause mortality and worse outcomes after liver transplantation, signaling a continued active area of research and an opportunity for improved management [17,18].

Considering the deleterious effects of insulin resistance and metabolic syndrome on body composition coupled with enhanced rates of muscle breakdown in patients with advanced liver disease, patients with MASLD are at high risk of sarcopenia, especially as the disease progresses [15,16,17,18].

## 3. Pathophysiological Considerations of Sarcopenia in MASLD

MASLD is defined by the dysregulation of metabolic pathways, which may be implicated in the enhancement of sarcopenia [19] (Figure 1). These mechanisms of disease that tie MASLD to loss of lean muscle mass include insulin resistance, lipogenesis, chronic inflammation, physical inactivity, and vitamin D deficiency [15,16,17,18,19].

### 3.1. Insulin Resistance and Lipogenesis

Insulin resistance (IR) occurs when a greater than normal amount of insulin is required for an appropriate physiologic response to serum glucose. This abnormal response to insulin plays an important role in pathologic states such as type 2 diabetes, obesity, and MASLD, and may play a role in the development of sarcopenia. The mechanism of the insulin signaling cascade is not within the scope of this review, but the end effects are important. Insulin activates glucose transport, glycogen synthesis, and lipogenesis, and downregulation of gluconeogenesis and lipolysis [19]. The hallmark of MASLD is fat deposition in hepatocytes, which has direct hepatotoxic effects. IR leads to increased lipolysis and release of free fatty acids (FFAs) from adipose tissue, [20] in turn leading to increased triacylglycerol (TAG) accumulation in the liver through esterification of the FFAs. A study by Donnelly et al. demonstrated the importance of IR and peripheral fat lipolysis in the pathogenesis of hepatic steatosis, showing that approximately 59% of the hepatic triglyceride deposition was derived from peripheral adipose tissue, which then made its way to the liver as non-esterified fatty acids (NEFAs) [21]. IR also leads to hyperinsulinemia and hyperglycemia promoting lipogenesis, the de novo synthesis of fatty acids, and inhibition of beta oxidation and breakdown of fatty acids, further contributing to MASLD [22].

Insulin resistance may have a direct sarcopenic effect on skeletal muscle, the largest organ system in the body, as well. Skeletal muscle has a significant role in glucose homeostasis via insulin-mediated glucose uptake through GLUT-4 glucose transporter [23]. In skeletal muscle, insulin primarily has anabolic effects via the activation of phosphatidylinositol 3-kinase (PI3K) and regulation of the mammalian target of rapamycin (mTOR). The overall effects of insulin include inhibition of muscle atrophy and stimulation of protein synthesis [24]. Therefore, insulin resistance can lead to inhibition of the anabolic effects of insulin resulting in muscle atrophy and sarcopenia, which may be pronounced in patients with advanced steatotic liver disease [25,26].

### 3.2. Obesity and Inflammation

There is a worldwide trend toward increased prevalence of obesity, which now affects one in three adults globally and is considered the most common risk factor for MASLD [27]. Obesity is defined by elevated BMI (>30 kg/meters^2^ (kg/m^2^)) and is associated with a chronic, low-grade inflammatory state that enhances IR and predisposes to body compositional changes that favor increased adiposity and a loss of lean body mass [28,29]. Obesity has multiple manifestations such as hypertrophy, hyperplasia, and activation of metabolically active cells called adipocytes, leading to further enhancement of the chronic inflammatory state via secretion of pro-inflammatory cytokines such as TNF alpha, interleukin-6, leptin, and adiponectin. Other less-studied and recently discovered cytokines include resistin, visfatin, retinol-binding protein, and chemerin. Secretion of these cytokines leads to the activation of inflammatory cascades driven by macrocytes and lymphocytes, which induces adipose tissue inflammation, decreased muscle protein synthesis, and loss of muscle mass and function leading to sarcopenia [30,31,32,33]. The phenomenon of obesity and sarcopenia existing together has been termed sarcopenic obesity and is a driver of morbidity and mortality associated with metabolic-associated diseases [24,30,32].

The balance of these cytokines is dysregulated with MASLD progression as well, leading to deleterious effects. Leptin secretion is upregulated in states with increased adiposity such as MASLD. Initially, leptin is thought to have anti-steatotic effects, but with the progression of MASLD, leptin develops proinflammatory and fibrogenic properties and may enhance disease progression and associated complications [34]. In contrast, adiponectin maintains an anti-inflammatory role throughout MASLD progression and has protective hepatocyte effects such as anti-steatotic, anti-fibrotic, and anti-apoptotic properties [34,35,36]. Levels of adiponectin, however, vary across steatotic liver disease states, likely related to hepatic clearance of the hormone [34,37,38]. These dysregulated pathways and cycling serum cytokine levels further alter body composition.

### 3.3. Physical Inactivity

The benefits of physical activity in metabolic-related chronic diseases are well known. The primary intervention in MASLD is weight loss through a hypocaloric diet and increased physical activity. Exercise is a type of physical activity that is purposeful and has a goal of improvement in health, with aerobic exercise promoting increased cardiovascular endurance, and strength training enhancing lean body mass development. Additionally, both types of exercise are associated with a reduction in metabolic dysfunction and may be implicated in reduced hepatic and visceral lipogenesis [39,40]. With increasingly sedentary lifestyles, there has been an increased prevalence of both obesity and MASLD [41]. Sedentary behavior has been linked to a decrease in the anti-lipolytic activity of insulin leading to enhanced IR [42]. In skeletal muscle, physical inactivity has catabolic effects and can lead to decreased lean muscle mass, while exercise is known to induce insulin sensitization and promote muscle protein synthesis, reducing the risk of sarcopenia [25]. The combination of overfeeding and physical inactivity can lead to obesity and disuse muscle atrophy, which is an increasingly important treatment consideration in patients with MASLD [43].

### 3.4. Vitamin D Deficiency

Vitamin D has numerous roles in the body. It acts via the vitamin D receptor (VDR) and has downstream effects in important organ systems including the liver and skeletal muscle [25,44]. The association between vitamin D, MASLD, and sarcopenia has been previously studied and has important clinical considerations. Research in rat models has shown that vitamin D exhibits anti-inflammatory properties and low vitamin D levels can cause progression of MASLD complications in genetically primed individuals [45,46]. Vitamin D also plays an anti-fibrotic role via the inhibition of stellate cells and profibrotic factors [44]. As previously discussed, IR is intimately tied to the development and progression of MASLD and studies have shown that vitamin D deficiency can also enhance IR, which may further portend poor outcomes related to enhanced disease activity and associated complications [44,46]. Longitudinal cohort studies have shown a correlation between low vitamin D levels and decreased muscle strength, a component of sarcopenia [47]. It is also thought that vitamin D deficiency can directly cause sarcopenia due to decreased oxygen consumption and increased levels of reactive oxygen species causing mitochondrial dysfunction and potentiating the above-described inflammatory cascade [48,49].

## 4. Evaluation of Sarcopenia in Patients with MASLD

There are several influential organizations with criteria for diagnosing sarcopenia. One of the most widely used guidelines is set by EWGSOP2, which has been previously reviewed and includes the combination of muscle mass loss with associated reduction in strength and function. Patients with MASLD, especially when it is complicated by advanced liver disease and fibrosis, are at enhanced risk of sarcopenia as previously discussed. Evaluation of sarcopenia is important in this population and includes both assessment of muscle strength and muscle mass (Table 4).

Muscle strength can be evaluated with maneuvers such as the handgrip strength and chair stand test. Handgrip testing has been shown to correlate well with overall muscle strength and is an easy test to perform in the outpatient setting [13,50]. Due to its ease in assessing muscle strength, it is a commonly used test limited only by equipment availability. The test involves squeezing a hydraulic dynamometer and recording the highest reading generated [51]. If equipment is not available, the chair stand test can be used, which is a good indicator of quadriceps strength [13]. In this test, the patient performs consecutive sequences of rising from a chair without the use of their arms in a set period, with more repetitions inferring higher muscle strength and function.

Muscle mass alternatively can be assessed by imaging such as computed tomography (CT), magnetic resonance imaging (MRI), or dual-energy x-ray absorptiometry (DEXA). Bioelectrical impedance analysis (BIA) and anthropomorphic measurements can also be used to define body composition and trend these values over time in the outpatient setting. MRI is shown to be highly accurate in the assessment of skeletal muscle mass and is considered the gold standard alongside CT [13,52]. MRI and CT can quantify the cross-sectional diameter of muscle and fat mass located at the T3 vertebrae or the mid-thigh region, which can be compared with age-specific controls and trended over time [53]. CT or MRI imaging of the mid-thigh has also been shown to be an accurate predictor of total body skeletal muscle mass [13]. Limitations of these imaging modalities include the need for specialized personnel for use, access, radiation exposure, and cost [50,52].

DEXA is a quick and low-cost imaging modality associated with low levels of radiation. An advantage of DEXA is that it can estimate the appendicular skeletal muscle (ASM). It can also calculate additional body composition indices including the presence of bone, fat-containing soft tissue, and lean soft tissue allowing for an estimation of appendicular skeletal lean mass, which is an important parameter of sarcopenia [52]. Disadvantages of DEXA include variability of assessment due to hydration status and water retention [13,52]. Assessment of muscle quality is also limited with this modality, as it is unable to distinguish intramuscular fat deposition [50,52]. DEXA is cost-effective for trending lean muscle mass, and wide availability has increased its use in clinical practice.

BIA is a method that uses electrical conductivity to estimate fat and lean body mass [13,50]. A study comparing BIA and DEXA concluded that BIA had 80% sensitivity and 90% specificity for detecting appendicular lean mass, compared with DEXA [54], further proving its potential in sarcopenia detection. This method is affordable and easy to use. Variability in results can be seen based on hydration status and fluid retention in certain disease states [52]. Besides BIA, muscle ultrasonography is a non-invasive test useful in assessing sarcopenia in the outpatient clinic. Muscle ultrasound, especially of the quadriceps muscle, is reliable in providing information on cross-sectional area and muscle thickness at a low cost [52]. Anthropomorphic measurements can be used as a tool to screen at-risk individuals if other imaging modalities are not available or cost-prohibitive [13,50]. This test can be administered in the clinic by conducting a measurement of the calf circumference or mid-arm muscle circumference, which can be used to grossly assess nutritional status and screen for sarcopenia [54,55].

Physical performance is another important index to assess in patients with advanced liver disease. This can be tested easily and conveniently via gait speed and/or the six-minute walk test. The 4 m gait speed test involves witnessing patients walk four meters while the clinician evaluates their speed. EWGSOP2 has used a speed of less than 0.8 m per second as an indicator of poor physical performance and sarcopenia [13]. The six-minute walk test is used to assess a patient’s aerobic capacity, therefore also evaluating physical performance [50]. Some other commonly used tests are the timed up and go test and the short physical performance battery test.

### Special Consideration in Advanced Liver Disease and Cirrhosis

Many tests that are useful in the diagnosis of sarcopenia are not as reliable in patients with decompensated cirrhosis, who may have variability in volume distribution intracellularly and extracellularly (Table 5). This accumulation of fluid can lead to ascites and edema and may decrease the efficacy of specific testing modalities. As discussed earlier, DEXA calculates different body compositions and cannot differentiate between muscle mass and edema in the setting of volume overload, which leads to an underestimation of sarcopenia [55]. BIA and anthropometric measurement have also been shown to be altered with health conditions that are complicated by fluid retention, limiting use in decompensated cirrhosis [52].

The liver frailty index (LFI) is a validated, easily calculated tool that assesses muscle strength and function in patients with end-stage liver disease. It relies on a combination of three physical performance tests including (1) handgrip strength, (2) chair stands, and (3) balancing exercises. Reduction in LFI score has been correlated with enhanced mortality in patients with cirrhosis waiting for liver transplantation and a predictor of all-cause mortality not related to underlying liver disease [56].

## 5. Management of Sarcopenia in Patients with MASLD

Nutritional and physical activity optimization is the primary intervention for managing and reducing sarcopenia progression, which becomes even more imperative in patients with MASLD. Early screening and appropriate intervention are key to ensuring the implementation of early treatment to halt the progression of sarcopenia in this at-risk patient population.

### 5.1. Nutritional Interventions

Nutritional interventions among patients with sarcopenia and MASLD should begin with an accurate quantification of calorie and protein intake in comparison with calculated need. Dietary protein is important in the synthesis and preservation of muscle mass and decreased dietary protein intake can lead to a catabolic state, promoting muscle breakdown to meet bodily protein requirements. The commonly accepted recommended dietary allowance (RDA) for the general adult population is 0.8 g (g) per kilogram (kg) of body weight, but these requirements are increased in sarcopenia as well as in associated chronic diseases such as liver and pulmonary disease, cancer, and in patients with advanced age. The combination of sarcopenia and advanced liver disease necessitates important consideration of protein requirement calculations and provision. Anabolic resistance, or the blunting of the protein synthesis response to normal stimuli such as dietary protein and exercise, drives further protein needs in this population [57]. Studies have shown that increased levels of dietary protein intake can lead to the preservation of handgrip strength and lean muscle mass [58,59,60]. Therefore, it is recommended that older adults, frail older adults, and individuals with acute or chronic diseases such as advanced liver disease, increase their dietary protein intake to 1.2–1.5 g per kg of body weight [61].

To help achieve higher protein goals, the ICFSR recommends the use of protein supplementation [62]. Insufficient caloric intake leads to a deficit in daily energy requirements. This can cause a catabolic state and breakdown of fat and muscle to provide energy leading to frailty or sarcopenia [63]. Multiple studies have shown that elderly adults with sarcopenia have decreased protein intake in addition to decreased intake of carbohydrates and fats leading to lower levels of calories and energy [64,65]. This variation in macronutrient intake may be altered in patients with MASLD, where obesity and metabolic syndrome are driven by high caloric intake, but low dietary quality, further underlining the need for early nutrition intervention.

In addition to total protein intake, the protein’s composition is also an important consideration in patients with MASLD. Branched-chain amino acids (BCAAs), such as leucine and isoleucine, are important building blocks in protein synthesis and play a role in muscle growth. In the elderly, anabolic resistance leads to decreased amino acid absorption and uptake in the muscles, leading to decreased muscle protein synthesis [66]. In a recent study, short-term BCAA supplementation led to increased physical performance, muscle mass, and strength in both elderly and sarcopenic patients [67]. When BCAA supplementation was stopped, a decline in markers of physical performance and muscle mass/strength was seen. Another study demonstrated the beneficial effects of leucine administration on walking time and lean mass index, which are two important sarcopenic criteria [68]. A metabolite of the essential amino acid leucine, β-hydroxy-β-methylbutyrate (HMB), has also been studied pertaining to its anabolic effects on skeletal muscle by increasing protein synthesis and decreasing proteolysis. A meta-analysis concluded that HMB supplementation led to the preservation of muscle mass compared with control groups in older adults [69]. Overall, studies have shown that protein and BCAA supplementation may play an important role in patients with sarcopenia by attenuating declines in muscle mass and strength.

Vitamin D deficiency plays an important role in sarcopenia, but vitamin D supplementation research has been inconclusive. One study concluded that adults with serum vitamin D levels < 25 nmol/L were more likely to develop sarcopenia compared with adults with vitamin D levels > 50 [70]. On the other hand, a randomized controlled trial concluded there was no significant difference in lean mass and leg press strength in two groups who were either administered placebo pills or vitamin D supplementation [71]. Currently, vitamin D supplementation to ensure that levels stay within the normal range is recommended although this is an area in which further data are needed to help with guidelines.

### 5.2. Exercise and Physical Activity

The ICFSR has stated that physical activity and resistance-based training (RT) should be first-line therapy when managing sarcopenia due to their effectiveness in improving muscle strength, muscle mass, and physical function [62]. The effectiveness of RT alone without protein supplementation is seen in many randomized control trials (RCTs). A small RCT evaluated the effectiveness of body weight-based and elastic band RT in sarcopenic women greater than 65 years old over 16 weeks. The conclusion was that women in the RT group had statistically significant increases in grip strength, gait speed, and isometric muscle strength compared with the placebo group [72]. Several other RCTs that utilized RT in patients with sarcopenia had statistically significant changes in handgrip strength, skeletal muscle mass index, body composition, and physical performance [73,74,75]. Another exercise intervention known as multimodal exercise combines RT, aerobic exercises, and balance training. Multimodal regimens have been shown to be beneficial for muscle strength and physical performance [76,77]. A combination of physical activity and supplementation has also shown beneficial results in sarcopenia. The use of supplements such as whey and casein protein, creatine, vitamin D, and fatty acids with exercise can lead to improvement in muscle mass and strength in adults at risk for sarcopenia [78,79].

### 5.3. Special Considerations in Advanced Liver Disease and Cirrhosis

Cirrhosis is a hyper-metabolic state with increased daily energy and protein requirements [80]. In advanced liver disease, the total energy expenditure (TEE) needs to be calculated to ensure patients are receiving appropriate calorie and energy intake. Currently, it is recommended that patients with advanced liver disease consume at least 35 kilocalories (kcal) per kg of body weight per day [81,82]. Sarcopenic obesity is common in patients with cirrhosis, especially in the setting of MASLD/MASH, and for these patients, caloric requirements are tailored to 25–35 kcal per kg of body weight per day for a BMI of 30–40 kg/m^2^ and 20–25 g per kg of body weight per day for a BMI > 40 kg/m^2^ [81,83].

Meal timing is also important for patients with advanced liver disease, and it is recommended that they should not undergo prolonged fasting. To prevent fasting, patients should have an early breakfast and late evening snack with frequent snacks in between, ideally every 3–4 h [80,81,82]. Late evening snacks are also important in cirrhosis to help minimize the deleterious effects of the catabolic state that ensues during fasting overnight and may further worsen sarcopenia. Plank et al. concluded that a nutrient-dense nighttime snack results in increased total body protein and lean tissue in patients with cirrhosis [84]. Other studies have shown late evening snacks have the potential to overcome anabolic resistance, proteolysis, and malnutrition leading to possible improvements in overall prognosis and quality of life [85,86,87,88].

Patients with cirrhosis are often incorrectly instructed to reduce protein intake to minimize hepatic encephalopathy (HE). Research has shown that these patients should not have protein restrictions and there is not an increased risk of HE with protein consumption [89,90,91]. The European Association for the Study of the Liver (EASL) recommends that patients with HE should not have restrictions on their dietary protein consumption, encouraging vegetarian and dairy sources of protein intake [82]. A daily protein intake between 1.2 and 1.5 g/kg of body weight is recommended for patients with cirrhosis to help ensure adequate intake and preservation of muscle mass [81,82]. Research has also been ongoing regarding the safest type of protein intake for patients with advanced liver disease. While there are no clear guidelines recommending the use of one dietary source over the other in this population, some small-scale studies have shown that non-animal-based protein sources may have some benefit in HE [92,93].

The BCAAs leucine, isoleucine, and valine are thought to have a role in the development of sarcopenia in advanced liver disease patients. The hyper-catabolic state in cirrhosis leads to the use of BCAAs as energy sources, leading to decreased levels in the body [80,94]. Data regarding the use of BCAAs is ambiguous. A meta-analysis concluded that oral BCAAs had a positive impact on HE but did not affect mortality or quality of life [95]. A study comparing a high protein and fiber diet plus oral BCAAs with a control group with only high protein and fiber showed increased muscle mass in the experimental group [96]. Although BCAA supplementation can be considered in patients who cannot tolerate meat protein or meet daily goals, long-term supplementation is not recommended [82]. Instead, it is recommended that patients meet their daily requirements through the utilization of multiple protein sources [81].

Malnutrition and sarcopenia are common in patients awaiting liver transplants (LTs) and are associated with longer hospital stays, increased risks of infection, and mortality pre- and post-transplant [97,98,99,100]. Several studies have shown the importance of preoperative muscle mass evaluation for prognostication of LT survival [100,101,102]. While LT reverses the complications of cirrhosis, it does not resolve sarcopenia. Studies have shown that in the post-LT phase, sarcopenia persists and continues to worsen [103,104]. Another study concluded that 26% of cirrhosis patients without sarcopenia pre-transplant had developed sarcopenia post-LT [105]. The persistence of sarcopenia post-LT can in part be due to the chronic use of immunosuppressant drugs, such as corticosteroids, which can lead to proteolysis and muscle wasting. Another potential cause of post-LT sarcopenia is thought to be the persistence of the hyper-metabolic state of cirrhosis into the post-transplant phases [106,107]. There has been research evaluating the effects of physical exercise rehab on pre- and post-LT patients, with results showing improvement in overall functional capacity and better control of comorbid metabolic diseases found in this patient population [108,109,110]. Improvements in functional status have also been used as a predictor for mortality in post-LT pediatric patients [111]. While sarcopenia is a known risk factor for mortality in LT patients, further research to attenuate its effects in this patient population is needed.

Lastly, vitamin D deficiency is commonly seen in patients with advanced liver disease and while the data are not equivocal, it is recommended that vitamin D be appropriately supplemented. Currently, the EASL recommends that patients with cirrhosis and vitamin D deficiency should receive oral supplementation to reach vitamin D levels > 30 ng/mL [82].

## 6. Conclusions

MASLD continues to be a growing concern with the prevalence of both metabolic syndrome and obesity increasing worldwide. Sarcopenia is increasingly associated with advanced liver disease and portends higher morbidity and mortality in this patient population. Exercise and nutrition are two interventions to reduce the progression of sarcopenia and MASLD and should be explored early in a patient’s clinical course in patients who screen at higher risk of advanced disease. Patients with advanced liver disease should be regularly screened for sarcopenia with EWGSOP2 guidelines, which assess physical performance, muscle mass, and strength and are commonly used for diagnosis. When diagnosing sarcopenia in decompensated cirrhosis, it is important to remember the limitations of specific tests used for muscle mass screening due to the accumulation of extracellular fluid (Table 5). If formally diagnosed with sarcopenia, a multi-pronged intervention that includes nutritional counseling with macronutrient manipulation favoring enhanced high-quality protein intake and a multimodal exercise regimen is recommended (Table 6). This ideally occurs with the support of multidisciplinary healthcare professionals including dietitians and physical therapists.

## Figures and Tables

**Figure 1 nutrients-16-00658-f001:**
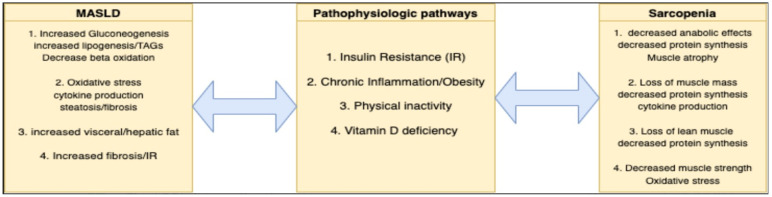
Relationship of pathophysiologic pathways in sarcopenia and MASLD.

**Table 1 nutrients-16-00658-t001:** Sarcopenia diagnostic criteria for strength.

Testing Modality	Cut-Off Points
Grip strength	Females: <16 kg Males: <27 kg
Chair stand	>15 (s) for five chair raises

**Table 2 nutrients-16-00658-t002:** Sarcopenia diagnostic criteria for muscle quality.

Testing Parameter	Cut-Off Points
ASM	Females: <15 kg Males: <20 kg
ASM/height^2^	Females: <5.5 kg/m^2^ Males: <7 kg/m^2^

**Table 3 nutrients-16-00658-t003:** Sarcopenia diagnostic criteria for performance.

Testing Modality	Cut-Off Points
4 m gait speed	≤0.8 m/s
400 m walk	Not able to complete ≥6 to complete
Short physical performance battery (SPPB)	Score: ≤8
Timed up and go test (TUG)	≥20 (s) to complete

**Table 4 nutrients-16-00658-t004:** Sarcopenia screening modalities and assessment criteria.

Sarcopenia Component	Screening Modality	Assessment Criteria
Muscle mass	CT/MRI	*Cross-sectional imaging of mid-thigh or L3 vertebra*
	DEXA	*Assessment of appendicular skeletal mass*
	BIA	*Electrical analysis of fat and lean body mass*
	Ultrasonography	*Cross-sectional area/muscle thickness*
	Anthropometry	*Measurement of calf/midarm circumference*
Muscle strength	Handgrip strength	*Measurement of strength with dynamometer*
	Chair stand	*Time required to stand from a seated position*
Physical performance	4 m gait speed	*Evaluation of speed*
	6 min walk	*Evaluation of aerobic capacity*

**Table 5 nutrients-16-00658-t005:** Evaluation of muscle mass screening modalities and limitations in advanced liver disease.

Testing Modality	Advantages	Limitations in Advanced Liver Disease
MRI	Highly accurate, low radiation	
CT	Highly accurate	
DEXA	Fast, low radiation, inexpensive	Fluid retention leads to underestimation of sarcopenia
BIA	Fast, no radiation, reproducible	Results affected by fluid retention and hydration status
Ultrasound	Fast, reproducible, no radiation	
Anthropometry	Fast, broadly available, inexpensive	Results affected by fluid retention

**Table 6 nutrients-16-00658-t006:** General dietary and exercise recommendations for patients with MASLD.

Calories	Requirements should be individualized for patients; general guidance is as follows: Non-obese (BMI < 30 kg/m^2^): 35 kcal/kg/day. BMI 30–40 kg/m^2^: 25–35 kcal/kg/day. BMI > 40 kg/m^2^: 20–25 kcal/kg/day.
2. Protein	Target intake of 1.2–1.5 g/kg/body weight per day. Utilization of multiple dietary sources for protein intake such as vegetables, dairy, and animal products. Utilization of protein supplements such as whey and casein to meet protein requirements.
3. BCAAs	Can be considered to meet protein requirements May reduce the risk of sarcopenia
4. Dietary habits	Well-balanced meals including breakfast, lunch, and dinner. Frequent snacks every 3–4 h between meals. High-density late-evening snack. Minimization of prolonged fasting periods.
5. Exercise	A regimen consisting of both aerobic physical activity and resistance-based training to enhance body composition and strength.

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
