# Peer review of "Implications of Protein and Sarcopenia in the Prognosis, Treatment, and Management of Metabolic Dysfunction-Associated Steatotic Liver Disease (MASLD)"

_nutrients, 2024, doi:10.3390/nu16050658_

Round 1

Reviewer 1 Report

Comments and Suggestions for Authors

A very timely and well written article- my only comments are the following-

1. line#75- HCC in MASLD can be present in those with and without cirrhosis- this should be corrected- PMID: 26196445

2. The obesity section did not flow as well as the other sections- I would suggest perhaps reorganizing it to reduce some redundancy.

3. I think a figure showing how the risk factors for MASLD and sarcopenia interact would be very helpful for the reader.

4. In line 387- "adequate" was misspelled which I am sure happened when using track changes.

  •  

  •  

Author Response

Dear Reviewer:

Thank you for taking the time to review our manuscript and providing valuable feedback. Below are the responses to your comments. We hope these corrections are adequate.

  1. Thank you for correcting the paper regarding our statement pertaining to HCC in MASLD patients, we used the study you included and made the appropriate edits.
  2.  The obesity section was re-organized which included removing redundant statements and rephrasing to help with improving the readability of the section.
  3. We created a figure for visual representation of the pathways linking MASLD and Sarcopenia.
  4. Fixed spelling errors in the paper included line 387.

Thank you for your feedback and consideration.

Reviewer 2 Report

Comments and Suggestions for Authors

Singh et al proposed an interesting paper concerning links between sarcopenia and MASLD.

Paper is well-written, not exhaustive in citations, but with the most recent generally.

I've got only some comments concerning revisions:

- In sarcopenia diagnosis, thresholds may be described or reported;

- A summary schema would be interesting for management since diagnosis to therapy;

- Authors speak about vitamin D role. They must add if there is a role of vitamin D substitution;

-Be careful, lines 322 and 326, with BCCA and BCAA.

Author Response

Dear Reviewer:

Thank you for taking the time to review our manuscript and providing valuable feedback. Below are the responses to your comments. We hope these corrections are adequate.

  1. In the section about sarcopenia we added three separate charts which comment on the parameters surrounding sarcopenia diagnosis. This involves thresholds pertaining to muscle strength, muscle quality and physical performance.
  2. Our conclusion was revised to include a summary which includes information about sarcopenia diagnosis and therapeutic considerations in advanced liver disease.
  3. We revised our statement regarding vitamin D supplementation by including information on current recommendations as per EASL.
  4. Fixed spelling errors as pointed out during the review.

Thank you for your feedback and consideration.
